# Dopamine D2-receptor blockade in humans disrupts the effect of effort on learning

Huw Jarvis[1]*, Oluwadamilola Obawede[1], Amy Q. Huynh[1], James P. Coxon[1], Mark A. Bellgrove[1], Trevor T.-J. Chong[1,2,3]*

**1** Turner Institute for Brain and Mental Health, School of Psychological Sciences, Monash University, Clayton, Australia, **2** Department of Neurology, Alfred Health, Melbourne, Australia, **3** Department of Clinical Neurosciences, St Vincent's Hospital, Melbourne, Australia

* huw.jarvis@monash.edu (HJ); trevor.chong@monash.edu (TT-JC)

## Abstract

Humans and other animals learn the value of candidate actions by interacting with their environment, which invariably requires the exertion of effort. Dopamine has been implicated in both effort and reward learning, but little is known about how these processes interact. In this double-blind study, healthy young adults ($N=42$) were randomized to receive either high-dose sulpiride (a post-synaptic D2-receptor antagonist) or placebo. Participants then completed a novel two-armed bandit task, in which they weighed the effort costs associated with each option against their expected rewards. Overall, learning accuracy was lower on sulpiride compared to placebo. Computational modeling revealed that this was driven by the capacity of effort to significantly modulate learning rates on placebo but, critically, not on sulpiride. Simulations showed that the capacity of effort to modulate learning rates plays an adaptive role by improving performance in agents whose learning would otherwise be compromised by low motivation. Together, these data provide causal evidence that dopamine supports the relationship between effort and learning, and reveal a novel role for dopamine in shaping how humans learn from the consequences of their actions.

## Introduction

The exertion of effort and the capacity to learn from our mistakes are both fundamental to daily life. Separate literatures have emphasized the importance of dopaminergic neurotransmission in supporting effort-based decisions [1–5], and reinforcement learning [6–13]. Despite often being studied independently, effort and learning are functionally and behaviorally related, and a topical question has been how to reconcile the roles of dopamine in both processes [14]. Data suggest that dopamine may support an interaction between effort and learning [15–17], but this proposal has not been directly tested in humans.

**Data availability statement:** Data and original modeling code used for the main analyses in this study are publicly available from the corresponding author's Github repository (https://github.com/huwjarv/sulpiride-effort-learning) and at the following link: https://doi.org/10.5281/zenodo.19245259.

**Funding:** M.A.B. is supported by grants from the National Health and Medical Research Council of Australia (App 2025415; App 2010899; https://www.nhmrc.gov.au). T.C. is supported by grants from the Australian Research Council (FT220100294; DP250102224; https://www.arc.gov.au). J.C. is supported by an Australian Research Council Future Fellowship (FT230100656; https://www.arc.gov.au). H.J. was supported by a Research Training Program (RTP) scholarship funded by the Australian Government Department of Education (CHESSN 3337449987; https://www.education.gov.au). The funders played no role in study design, data collection and analysis, decision to publish, or the preparation of the manuscript.

**Competing interests:** The authors have declared that no competing interests exist.

**Abbreviations:** AIC, Akaike Information Criterion; AW, Akaike weights; BBI, Bang Blinding Index; MVC, maximum voluntary contraction; PD, Parkinson's disease; RPEs, reward prediction errors.

Dopamine is known to play a causal role in decisions about whether to engage in effortful actions [18–20], and in the exertion of effort itself [1,2,5]. Humans and other animals generally consider effort to be aversive [18,21–23], such that the prospect of exerting effort reduces (or 'discounts') the subjective value of rewards on offer [24–26]. Traditional models of basal ganglia function stipulate that D2 receptors are primarily involved in movement inhibition [27,28], but more recent work has challenged this view. For example, D2 signaling is also linked to increased motor activation [29,30], and blockade of D2-receptors can heighten the aversiveness of effort [31].

A separate body of literature has implicated dopamine in the capacity to learn from reward [32–34]. Most notably, the firing rates of dopaminergic neurons convey the reward prediction errors (RPEs) that drive reinforcement learning [35–37]. Canonical models of reinforcement learning propose that individuals update their expectations by scaling the magnitude of the RPE as a function of a learning rate parameter [32], and previous work has shown that individuals tend to learn more quickly from positive relative to negative outcomes [38–41]. Disrupting dopaminergic signaling has been shown to impede reward-based learning [42–45], with D2-receptor blockade in humans appearing to modulate the relative efficiency with which individuals learn from rewards and punishments [44,46].

Together, these previous studies demonstrate an important role for dopamine in both the execution of effortful actions and the capacity to learn from reward. However, these two functions are typically studied in isolation, and an important question is how to reconcile these seemingly disparate functions of dopamine within a single framework [14,15,33]. There is growing evidence of a close behavioral and neurophysiological relationship between effort and learning. For example, exerting greater effort in a reward learning task augments the effect of positive RPEs, and attenuates the effect of negative RPEs [41]. Furthermore, studies in nonhuman animals suggest that effort can amplify dopaminergic activity associated with positive outcomes [47], and attenuate activity associated with negative outcomes [48]. These findings are consistent with the notion that the neural mechanisms underpinning reward-guided effort and learning are closely related, but a causal role for dopamine is yet to be established. An outstanding question, therefore, is whether intact dopamine signaling is necessary to support the effect of effort on learning in humans—and, if so, why.

In this study, healthy human volunteers completed a novel two-armed bandit paradigm, in which they had to exert high or low levels of physical force to register their responses [41]. We compared performance in participants on a dopamine D2-receptor antagonist (sulpiride, 800 mg; $n = 23$) against a separate group of participants on placebo ($n = 19$). First, in the placebo group, we aimed to replicate previous findings that effort modulates asymmetries in the efficiency of learning following rewarded and nonrewarded choices [41]. Next, we tested whether dopamine plays a causal role in maintaining this interaction by asking whether the effect of effort on learning was also present in the sulpiride group. Finally, we examined whether

PLOS Biology

dopamine plays an adaptive role in maintaining task performance by simulating the effects of preserved and disrupted D2 signaling on the interaction between effort and learning.

## Methods

### Ethics

The experimental procedures in this study were approved by the Human Research Ethics Committee of Monash University in Melbourne, Australia (Project ID: 26350). The study was conducted in accordance with the principles of the Declaration of Helsinki. All participants provided written informed consent before commencing the study.

### Participants

We recruited healthy adults aged between 18 and 40 years. Exclusion criteria were: females not on hormonal contraception; recent use of psychotropic medication or recreational drugs; personal history of neurological disease or psychiatric illness; and contraindication to the study drug. Four participants were excluded due to inconsistent behavior during calibration of the dynamometers, such that they exerted forces exceeding their nominal maximal voluntary contraction (MVC) by more than 10% during the experimental blocks. The final sample included 42 participants: 19 in the placebo group and 23 in the sulpiride group. Groups did not differ in terms of age, gender, and body mass index (all $p > .1$; Table 1).

### Study design

We used a double-blind, randomized design to compare a single 800 mg dose of sulpiride to placebo. At an 800 mg dose, sulpiride effectively blocks post-synaptic D2-receptors [49,50]. This study was planned and conducted as a within-subjects cross-over design, in which participants were administered either sulpiride or placebo (microcrystalline cellulose) in identical-looking capsules across two counterbalanced sessions separated by one week to ensure drug washout ($T_{1/2} = 8$ hours). However, post hoc analyses after unblinding at the conclusion of the study revealed strategic changes in learning following the first session, which resulted in significant differences in behavior between the two sessions (S1 Text; S1 Fig). Consequently, in this paper, we analyze the effect of drug versus placebo in the first session only using a between-subjects design [e.g., 13].

Participants were randomized according to a pre-determined schedule prepared by an independent researcher. After ingesting the corresponding capsule in each session, participants were seated for 2 hours to allow for sulpiride to reach peak plasma concentrations before commencing the practice blocks [51–53]. We took baseline and hourly measures of heart rate and blood pressure, and reports of subjective state using a digital version of the Bond and Lader Visual Analogue Scales [54].

### Drug reactions and blinding

Sulpiride did not alter heart rate or blood pressure relative to placebo (S1 Text; S2 Fig). There were no effects of sulpiride versus placebo on subjective reports of feeling strong/feeble, well-coordinated/clumsy, or lethargic/energetic, nor on

**Table 1. Group characteristics.**

|  | Placebo Group | Sulpiride Group | Comparison |
|---|---|---|---|
| *N* | 19 | 23 | – |
| Sex (male:female) | 10:9 | 17:6 | $\chi^2 = 1.23$, $p = .27$ |
| Age (years, mean ± SD) | 22.37 ± 3.3 | 24.43 ± 4.37 | $t(40) = -1.7$, $p = .1$ |
| Body Mass Index (kg/m², mean ± SD) | 23.38 ± 4.25 | 25.92 ± 5.88 | $t(40) = -1.57$, $p = .12$ |

aggregated factors of alertness, contentedness, and calmness. Small differences in feeling alert/drowsy did not survive correction for multiple comparisons (S1 Text; S3 Fig). No participant reported any adverse drug reactions during the testing session.

At the conclusion of testing, participants were asked whether they believed they had ingested the placebo, the active drug, or if they were unsure. These data were then used to calculate the Bang Blinding Index (BBI) for each drug group (1 = complete lack of blinding; 0 = perfect blinding; −1 = complete attribution of assignment to alternate group; [55]). Importantly, the BBI in both groups was close to zero, which reflects effective blinding in this study (placebo, BBI = 0 ± 0.17 (mean ± SEM), 95% CI = [−0.33, 0.33]; sulpiride, BBI = 0.26 ± 0.15, 95% CI = [−0.04, 0.56]; S1 Text). In addition, a supplementary analysis demonstrated that behavior was not significantly related to whether participants were able to correctly guess their assigned drug group (S1 Text).

## Behavioral paradigm

To examine the effects of effort on choice and learning, we used a reward learning paradigm involving the exertion of effort [Experiment 2 in 41]. Specifically, participants in both drug groups completed a two-armed bandit task in which they were required to register their responses by applying pre-specified levels of physical force ('low' or 'high') to a pair of hand-held dynamometers (SS25LA, BIOPAC Systems, USA). Target force levels were standardized for each participant as proportions of their maximal voluntary contraction (MVC; low = 5%; high = 44%). MVCs for each hand were defined at the beginning of the experiment as the maximum force generated from three ballistic contractions with the corresponding dynamometer.

The experimental task consisted of two blocks of 150 trials. On each trial, participants were presented with a pair of abstract stimuli (fractals) on the left and right of the screen, and were required to select which was more rewarding based on probabilistic feedback received on previous trials (Fig 1A). The left/right location of each stimulus was randomized on every trial. On any given trial, both stimuli could have a high probability of being rewarded ($P = 0.7$); both could have a low probability ($P = 0.3$); or one could be superior to the other ($P = 0.7$ versus $P = 0.3$). Stimulus-reward contingencies changed after every 12 or 24 trials according to a pseudorandomised sequence, and contingency changes were not signaled to participants (Fig 1B). On rewarded trials, a 'smiley face' was presented for 0.5 s accompanied by a positively-valenced auditory tone ('cash register' sound effect). On nonrewarded trials, a 'sad face' was presented for 0.5 s with a negatively-valenced tone ('wrong answer buzzer' sound effect). Participants had a maximum of 2 s to register a response on each trial, otherwise a 'Too slow!' message was displayed for 0.5 s and then the next trial began. Participants were incentivised by the opportunity to increase their remuneration based on their performance.

A critical feature of this task was that one of these two stimuli was always designated the 'low effort' stimulus, and could be selected by exerting only a negligible amount of force (>5% MVC). The other was designated the 'high effort' stimulus, and required a higher amount of force to be selected (>44% MVC). These stimulus-effort mappings remained constant for the duration of the experiment, and participants were explicitly informed about the identity of the low and high effort stimuli at the start of the testing session. To reinforce these stimulus-effort mappings, participants performed a preliminary block of 50 trials in which they were cued to generate the force corresponding to either the low or high effort stimulus (randomly determined). Participants then received binary feedback (correct versus incorrect) about whether they had generated the correct amount of force (5%–44% of MVC for the low effort stimulus, or >44% MVC for the high effort stimulus). In the subsequent experimental blocks, participants thus had to incorporate a consideration of both the effort required to select each stimulus, which was known in advance, as well as the expected reward, which had to be learned during the task.

The experiment was run in Psychtoolbox [56] implemented in Matlab version 9.4 (2018) [57], and presented on a monitor at a viewing distance of ~60 cm.

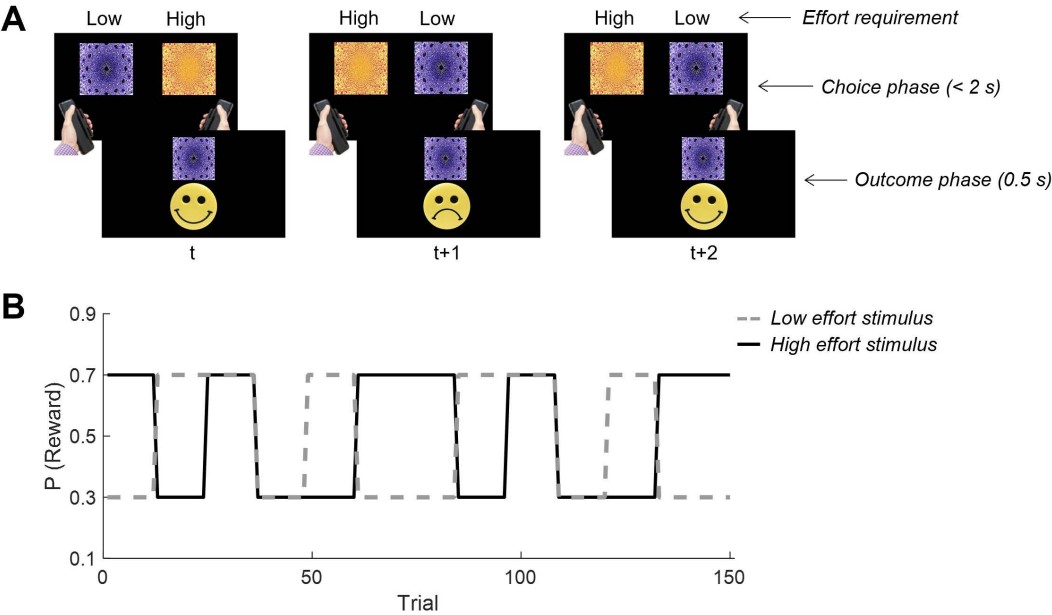

**Fig 1. Reward-based learning paradigm with effort manipulation. (A)** Participants made a series of choices between two abstract stimuli by applying physical force to a pair of hand-held dynamometers. One stimulus required a negligible amount of force to select (low effort stimulus; >5% MVC) and the other a greater amount of force (high effort stimulus; >44% MVC). Participants, therefore, had to balance an aversion to the high effort stimulus against their desire to maximize reward. **(B)** Stimulus-reward contingencies across a single block of 150 trials. The probability of reward upon selecting a given stimulus was either $P = 0.7$ or $P = 0.3$. These contingencies reversed every 12 or 24 trials, and reversals were not signaled to participants. Fractal images in panel (A) are by Optoskept, reproduced from https://commons.wikimedia.org under a Creative Commons Attribution 4.0 International license.

## Data analysis

**Statistical analysis.** Choice accuracy was defined as the proportion of trials on which the participant chose the stimulus associated with the higher probability of reward (when stimulus-reward contingencies were $P = 0.3$ versus $P = 0.7$). We also examined choice behavior with respect to win-stay and lose-switch strategies. Win-stay behavior was defined as the proportion of trials on which the participant selected the same choice stimulus again following a positive reward outcome. Lose-switch behavior was defined as the proportion of trials on which the participant switched to the alternative choice stimulus following a negative reward outcome. ANOVAs were fit with Type III sums of squares, and violations of sphericity were corrected using the Greenhouse-Geisser method [58]. All $t$ tests were two-tailed, and $p$-values were corrected for multiple comparisons using the Bonferroni method [59]. Missed choices were assessed using a two-sample rank-sum test due to nonnormality of these data. Statistical analysis was performed in Jamovi version 2.4 (2023) [60]. Plots were created in Matlab version 9.4 (2018) [57].

**Computational models of learning.** We investigated interactions between effort and learning on a trial-by-trial basis by considering a family of three computational models (M1-3) adapted from previous work [41]. All models shared a common structure that combined features of the classical Rescorla-Wagner model of reinforcement learning [61] with canonical models of effort discounting [25]. The core learning model stated that, on every trial ($t$), the value ($V$) of the chosen stimulus ($s$) is updated according to an RPE ($\delta$) (Eq. 1), which is the difference between the reward obtained ($r = 0$ or 1) and the reward expected based on the current stimulus value ($V$) (Eq. 2). The extent to which $V$ is updated by $\delta$ is determined by the learning rate ($\alpha$), which takes a value between 0 and 1.

$$V(s,\ t+1) = V(s,t) + \alpha \cdot \delta(t) \tag{1}$$

$$\delta(t) = r(t) - V(t) \tag{2}$$

To model the effect of effort on decision-making, we also incorporated an effort discounting function [25], which reduces the value associated with a stimulus by the amount of effort required to select it. We thus distinguish *reward* value ($V$), which participants can learn using trial-by-trial reward feedback, from *action* value ($V'$), which also accounts for the effort required to select the stimulus in question (Fig 2B). This effort cost enters the model as $E_c = 0.05$ for the low effort stimulus and $E_c = 0.44$ for the high effort stimulus, scaled by a subject-specific effort discounting parameter ($k$) to capture each individual's aversion to effort (Eq. 3).

$$V'(s, t) = V(s, t) - k \cdot E_c(s) \tag{3}$$

Finally, action values are converted into choice probabilities using a softmax function [62], in which the probability ($P$) of choosing a given stimulus ($s_1$) depends on its associated action value relative to that of the nonchosen stimulus ($s_2$; Eq. 4). An inverse temperature parameter ($\beta$) accounts for individual differences in choice stochasticity.

$$P(s_1, t) = \frac{1}{1 + \exp(-\beta \cdot [V'(s_1, t) - V'(s_2, t)])} \tag{4}$$

All three candidate models shared this common structure (Eq. 1–4), and differed only in how they estimated the learning rate ($\alpha$). To allow for the possibility that learning rate varies at the level of single trials, we first defined $\alpha$ as a sigmoidal function of a signal gain term ($G$) [41].

$$\alpha(t) = \frac{1}{1 + \exp(-G(t))} \tag{5}$$

We then let the specific contents of $G$ be the defining feature of each candidate model (Fig 2A).

**Baseline model (M1).** Model M1 reduces to a standard Rescorla–Wagner model, in which individuals learn equally well from positive and negative reward outcomes. In this model, $G$ is estimated directly via a subject-specific signal gain parameter ($\gamma$).

$$M1: \ G(t) = \gamma \tag{6}$$

**Dual Learning Rates model (M2).** Model M2 was similar to M1, but accounted for the possibility that learning rates differ following positive and negative reward outcomes. For example, a common finding in reward learning paradigms is that learning rates tend to be higher for positive relative to negative RPEs [8,38–40]. Model M2 accounts for this possibility by estimating the difference between positive and negative learning rates with an additional free parameter ($\varphi_N$). This model allows for learning to be more efficient from positive than negative RPEs ($\varphi_N > 0$), vice versa ($\varphi_N < 0$), or equally efficient from both ($\varphi_N = 0$).

$$M2: \ G(t) = \begin{cases} \gamma + \varphi_N, & \delta(t) > 0 \\ \gamma - \varphi_N, & \delta(t) < 0 \end{cases} \tag{7}$$

Note that, like M1, this model assumes that effort has no effect on learning rates.

**Effort Reinforcement model (M3).** Finally, based on our previous work, we included a model that accounts for the possibility that within-subject variations in learning rate are sensitive to the amount of effort invested in each choice [41].

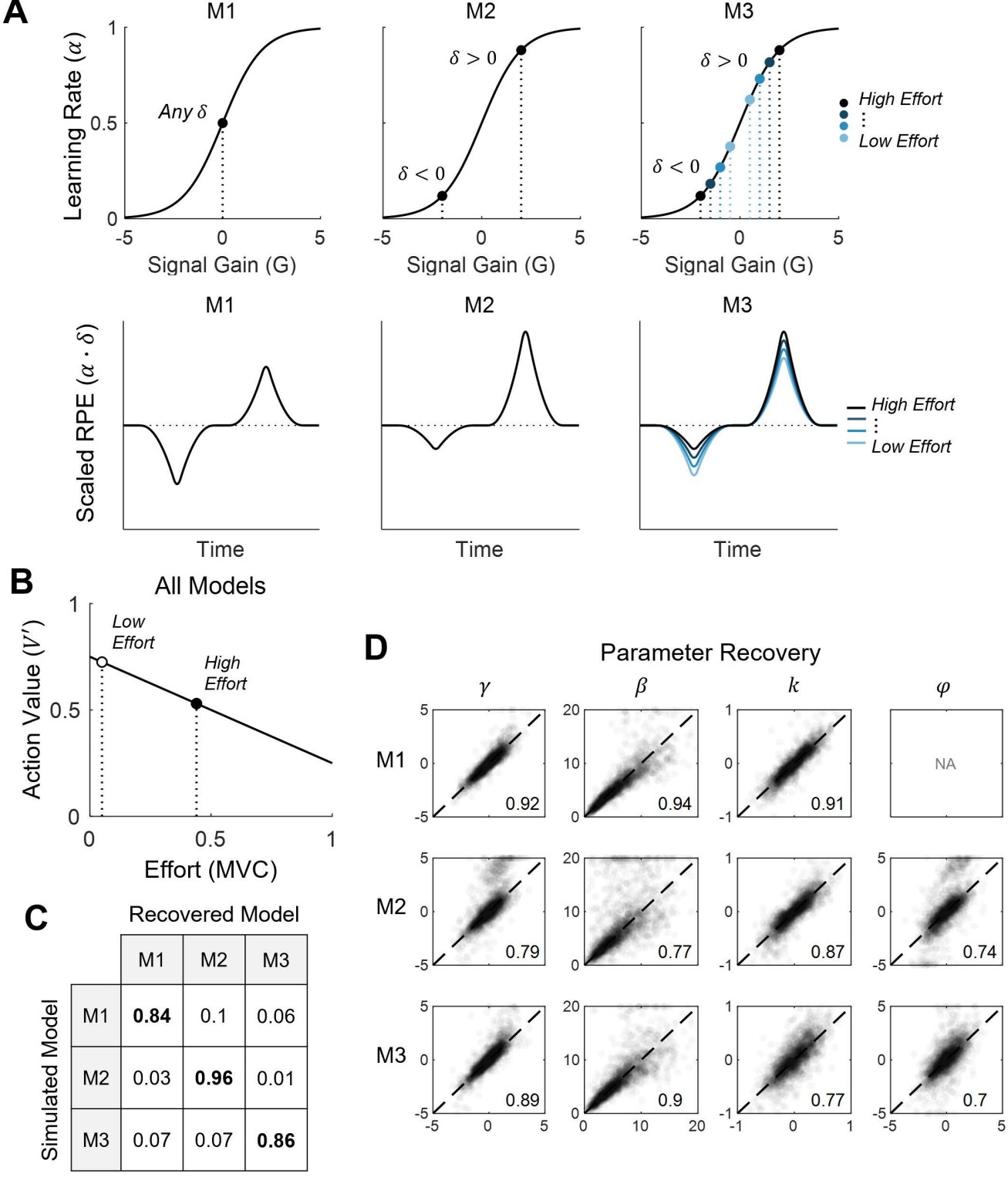

Fig 2. Candidate computational models of effort-based learning. (A) Schematic diagrams of candidate models (M1-3) depicting learning rates (α) (top row), and their predicted effects on RPEs (bottom row). M1 includes a single learning rate, M2 fits separate learning rates for positive and negative

RPEs, and M3 stipulates that learning rates are sensitive to trial-by-trial effort exertion ($E_x$). **(B)** The action value ($V'$) associated with a given stimulus is modeled as its expected reward value ($V$) discounted by the effort cost required to select it ($E_c$). Plot depicts $V = 0.75$, $k = 0.5$. **(C and D)** Candidate models were simulated 100 times, yielding model recovery accuracy ≥ 0.84 for all models **(C)**, and parameter reliability ≥ 0.7 for all parameters **(D)**.

This model is similar to M2 in assuming that learning rates are asymmetrical (i.e., differ following positive and negative reward outcomes). In M3, however, a $\varphi_E$ parameter scales the effect of effort exerted ($E_x$) on the current trial ($t$), defined as the peak amplitude of force exerted as a proportion of the participant's MVC. This model allows for the possibility that effort boosts learning from positive compared to negative RPEs ($\varphi_E > 0$), vice versa ($\varphi_E < 0$), or that effort has no effect on learning rates ($\varphi_E = 0$).

$$M3: \ G(t) = \begin{cases} \gamma + \varphi_E \cdot E_x(t), & \delta(t) > 0 \\ \gamma - \varphi_E \cdot E_x(t), & \delta(t) < 0 \end{cases} \tag{8}$$

Thus, in both M2 and M3, the $\varphi$ parameter captures the degree of learning rate asymmetry. The critical distinction is that, in M2, this parameter is not dependent on effort ($\varphi_N$). In contrast, in M3, this parameter captures the degree to which effort modulates learning by scaling the effect of effort on learning rate asymmetry ($\varphi_E$).

### Computational model fitting and comparison

Candidate models were fit to the observed choice data using maximum likelihood estimation. The best-fitting parameter values were estimated for each participant separately, using flat priors for all parameters. Model fits were compared based on the Akaike Information Criterion (AIC), which prevents overfitting by penalizing models that have a greater number of free parameters [63]. AIC scores were summed across participants to calculate overall model fits in each drug group. We also calculated Akaike weights (AW) to quantify the relative likelihood that the winning model best accounted for the observed data compared to others in the model space (Eq. 9):

$$AW(i) = \frac{\exp\left(-0.5 \cdot \Delta AIC(i)\right)}{\sum_{i=1}^{I} \exp\left(-0.5 \cdot \Delta AIC(i)\right)} \tag{9}$$

where $AW(i)$ is the AIC weight of model $i$; $\Delta AIC(i)$ is the difference in AIC between model $i$ and the winning model; and $I$ is the number of models in the space.

We confirmed that each candidate model was uniquely identifiable by conducting a model recovery analysis. We ran 100 simulations per model, each of which generated synthetic data from 20 learning agents completing the experiment. We then repeated our model fitting procedure on these synthetic data and calculated the proportion of simulations on which the true generative model was successfully recovered as the winning model (Fig 2C). We also quantified the reliability of the parameter estimates from each model as the median rank-order correlation (Spearman's ρ) between the true generative values and the recovered values across all 100 simulations (Fig 2D). Computational modeling was performed in Matlab version 9.4 (2018) [57].

## Results

### Sulpiride did not alter motor performance or effort preference relative to placebo

To test whether sulpiride affected motor capacity, we analyzed participants' force generation. First, we found no significant difference in MVCs recorded during the calibration phase between the sulpiride and placebo groups ($p = .54$; Fig 3A). We also confirmed that participants were able to successfully meet the required effort thresholds when registering responses in the experimental blocks (<0.7% of trials missed per participant on average), and found that this did not

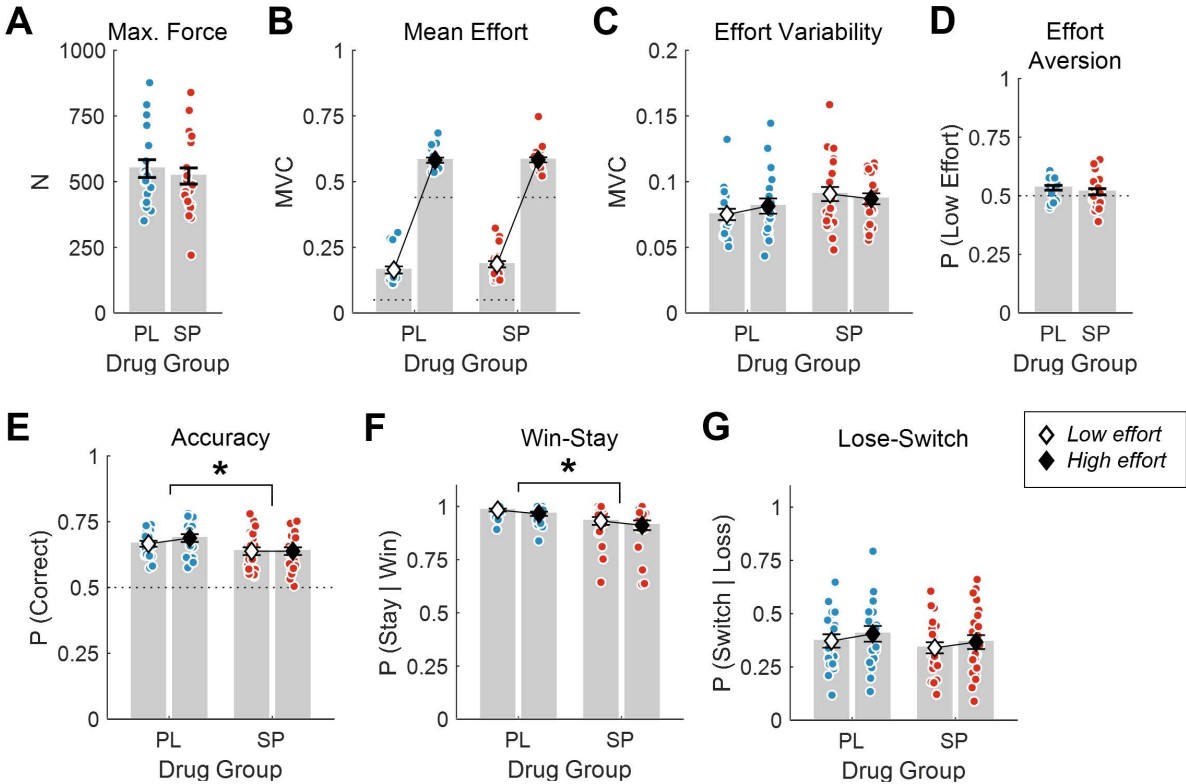

**Fig 3. Motor and learning performance in the sulpiride and placebo groups. (A-C)** Motor performance did not differ significantly between drug groups with respect to **(A)** MVCs, **(B)** mean effort exerted during the experimental blocks, or **(C)** standard deviation of effort during the experimental blocks. Dotted lines in **(B)** indicate minimum effort thresholds for the low and high effort stimulus, respectively. **(D)** Sulpiride did not affect preferences for low vs. high effort actions compared to placebo. **(E–G)** Learning performance differed between drug groups. Relative to placebo, participants on sulpiride exhibited **(E)** significantly lower choice accuracy (*p* = .042), and **(F)** were less likely to repeat rewarded choices (*p* = .029), but **(G)** were no different in their tendency to switch stimulus following non-rewarded choices (*p* = .4). In addition, across both drug groups, participants were less likely to repeat high effort choices than low effort choices, consistent with effort aversion (*p* ≤ .009; **F** and **G**). Accuracy in panel E is based on trials with reward contingencies of *P* = 0.7 vs. 0.3. Error bars depict the standard error of the mean. Statistical significance marks are shown for differences between drug groups. \**p* < .05; \*\**p* < .01; PL, placebo (*n* = 19); SP, sulpiride (*n* = 23). Data underlying this Figure can be found in S1 Data.

differ significantly between drug groups (z = −1.46, *p* = .15). Next, we used a two-way ANOVA to examine the average peak force exerted per response in the experimental blocks (defined as a proportion of each participant's MVC). Average effort exertion was compared between Drug groups (placebo, sulpiride) as a function of the Stimulus type (low effort, high effort). As expected, the main effect of Stimulus was significant, such that participants exerted significantly more effort when selecting the high versus low effort stimulus (ΔMVC = 0.41 ± 0.01 (mean ± SEM), t(40) = −36.8, *p* < .001). However, neither the main effect of Drug nor the two-way interaction was significant (both *p* ≥ .33; Fig 3B). We then ran an equivalent ANOVA testing differences in the standard deviation of peak force across responses (Effort SD). Importantly, the main effect of Drug group was not significant (ΔEffort SD = 0.01 ± 0.01 (mean ± SEM), F(1,40) = 3.36, *p* = .074; Fig 3C). The effect of Stimulus in this case was not significant (main and interaction both *p* ≥ .22; Fig 3C). Finally, we compared the overall effort bias of each group based on the tendency of participants to select the less effortful option. We found no significant difference in the proportion of trials on which the low versus high effort stimulus was chosen between the two groups (*p* = .33; Fig 3D).

In sum, we found no strong evidence that sulpiride affected either the motor capacity or the willingness of participants to produce forceful responses compared to placebo.

## Sulpiride impaired reward-based learning relative to placebo

Having established that sulpiride did not affect motor performance relative to placebo, we next tested whether sulpiride affected performance in the reward learning task. To do so, we first compared drug groups on choice accuracy, defined as the proportion of trials on which participants chose the more highly rewarded stimulus (on trials with unequal stimulus-reward contingencies). We used a two-way ANOVA to compare Accuracy between Drug groups (placebo, sulpiride) as a function of the Stimulus type (low effort, high effort). This revealed that, across both stimulus types, sulpiride reduced overall choice accuracy compared to placebo (Drug, $\Delta$Accuracy $= 0.04 \pm 0.02$, t(40) = 2.1, $p = .042$; Stimulus main effect and interaction, $p \geq .15$; Fig 3E).

We next ran analogous ANOVAs to investigate whether sulpiride altered the tendency to make Win-stay and Lose-switch choices, as more specific measures of reward-guided behavior. We found a significant main effect of Drug in the Win-stay analysis, indicating that the sulpiride group was less likely than the placebo group to repeat rewarded choices ($\Delta$Win-stay $= 0.05 \pm 0.02$, t(40) = 2.27, $p = .029$; Fig 3F). This analysis also revealed a main effect of Stimulus type, which reflected that, across both groups, participants were less likely to repeat rewarded high effort choices than rewarded low effort choices ($\Delta$Win-stay $= -0.02 \pm 0.01$, t(40) = $-2.75$, $p = .009$; Fig 3F). Notably, the Drug $\times$ Stimulus interaction was not significant ($p = .89$).

In the Lose-switch analysis, we found no significant difference between Drug groups ($p = .4$; Fig 3G). However, we again found a significant main effect of Stimulus type, which in this case indicated that participants were more likely to switch stimulus following nonrewarded high effort choices than nonrewarded low effort choices ($\Delta$Lose-switch $= 0.03 \pm 0.01$, t(40) = 2.9, $p = .006$; Fig 3G). Together with the Win-stay analysis, these effects of Stimulus type are indicative of effort aversion, such that participants were in general less likely to repeat high effort choices than low effort choices. Again, the Drug $\times$ Stimulus interaction effect was not significant, providing further evidence that sulpiride did not alter effort aversion in this study ($p = .75$).

In sum, these analyses reveal that overall reward-related performance was lower on sulpiride than placebo, and that this was associated with changes in win-stay, but not necessarily lose-switch, behavior.

## Effort modulated learning rates on placebo, but not sulpiride

We next turned to our central question of how effort modulates learning, and the role of the dopamine D2-receptor in maintaining this relationship. We compared three candidate reinforcement learning models (M1-3) in each drug group (see Methods section for details): a *Baseline model (M1)* with a single learning rate; a *Dual Learning Rates model (M2)* capturing different learning rates for positive and negative reward outcomes; and an *Effort Reinforcement model (M3)* in which learning rates are sensitive to trial-by-trial effort exertion [41].

In the placebo group, we found that choice behavior was best captured by the *Effort Reinforcement model (M3)*. This model yielded a superior fit compared to alternative models ($\Delta$ AIC $\geq 16.59$; Akaike weight $> 0.99$; Fig 4A–4C). Analysis of the key parameters in this model indicated that, as expected, the $k$ parameter was significantly greater than zero, indicating that the prospect of exerting effort discounted the value of candidate actions ($k = 0.24 \pm 0.06$, t(18) = 3.97, $p < .001$; Fig 5A). In addition, the $\varphi_E$ parameter was significantly greater than zero, indicating that the overall effect of effort in this group was to increase learning rate asymmetry, boosting learning rates following positive reward outcomes, and blunting learning rates following negative reward outcomes ($\varphi_E = 0.95 \pm 0.37$, t(18) = 2.59, $p = 0.019$; Fig 5A). To confirm that this model was not merely approximating dual learning rates independent of effort, we ran a permutation test in which we randomly shuffled the effort exerted across trials for each participant in the placebo group and re-fit the model. The empirical data provided a better fit than the permuted data on each of 1,000 permutations, confirming that learning rates in this group were sensitive to effort at the level of single trials (permuted $p < .001$).

We then repeated the same model comparison in the sulpiride group. Critically, choice behavior in these participants was best captured by the *Dual Learning Rates model (M2)*, in which learning rates are entirely independent of effort

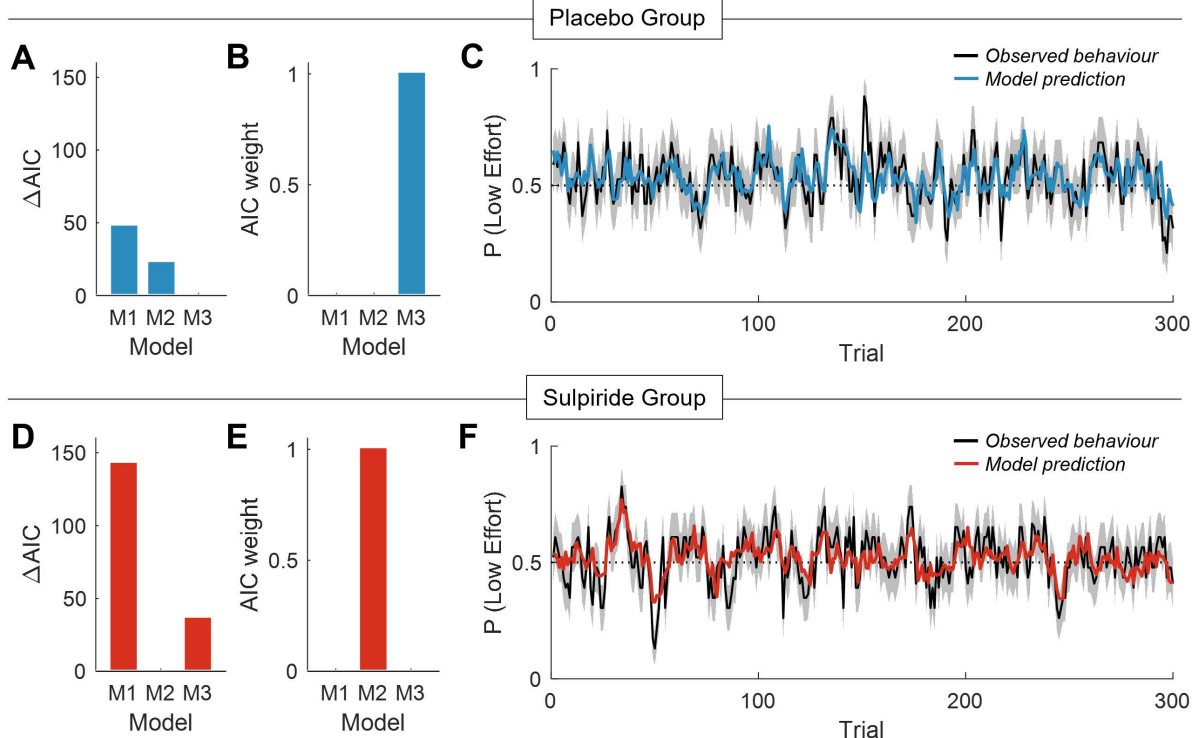

**Fig 4. Computational modeling results for the placebo and sulpiride groups. (A–C)** In the placebo group (top row, blue, $n = 19$), the empirical data were best explained by the *Effort Reinforcement model* (M3), which stipulates that learning rates are modulated by the exertion of effort. This model provided a superior fit to models M1 and M2 based on AIC scores **(A)** and Akaike weights **(B)**. **(C)** M3 also provided an accurate account of observed choices on a trial-by-trial basis. The black line shows the proportion of participants on each trial selecting the low effort stimulus (with standard error in gray), and the blue line shows the mean choice probabilities derived from M3. **(D–F)** In the sulpiride group (bottom row, red, $n = 23$), the empirical data were best explained by the *Dual Learning Rates model* (M2), in which effort has no effect on learning rates. This model was superior to the alternative models in this group on the basis of AIC scores **(D)** and AIC weights **(E)**, and provided an accurate account of observed behavior on sulpiride **(F)**. Data underlying this Figure can be found in S1 Data.

($\Delta$ AIC ≥ 34.41; Akaike weight > 0.99; Fig 4D–4F). As in the placebo group, $k$ values were significantly greater than zero, indicating that individuals found effort to be aversive ($k = 0.12 \pm 0.05$, t(22) = 2.48, $p = .021$; Fig 5C). Consistent with prior work, the $\varphi_N$ parameter was also positive, indicating that learning rates were higher following positive outcomes than negative outcomes ($\varphi_N = 2.06 \pm 0.46$, t(22) = 4.52, $p < .001$; Fig 5C).

Together, these results demonstrate that, across both the placebo and sulpiride groups, learning rates were higher for positive than negative reward outcomes. In the absence of dopaminergic blockade, effort reinforced learning by increasing learning rates for positive RPEs, and decreasing learning rates for negative RPEs [41]. Critically, however, this effect of effort on learning was absent in the sulpiride group, thus highlighting the importance of the dopamine D2-receptor in maintaining the effect of effort on reward learning.

## On placebo, effort-sensitive learning rates attenuate the detrimental effect of effort aversion on accuracy

These results raise the question of why the exertion of effort should modulate learning at all, and whether dopamine plays an adaptive role in maintaining this relationship. To address these questions, we first considered the placebo group, and asked whether the degree of effort discounting ($k$) was related to the degree to which effort modulated learning rate asymmetry ($\varphi_E$) in the winning model (M3). Importantly, we found that these parameters were positively correlated, such

   

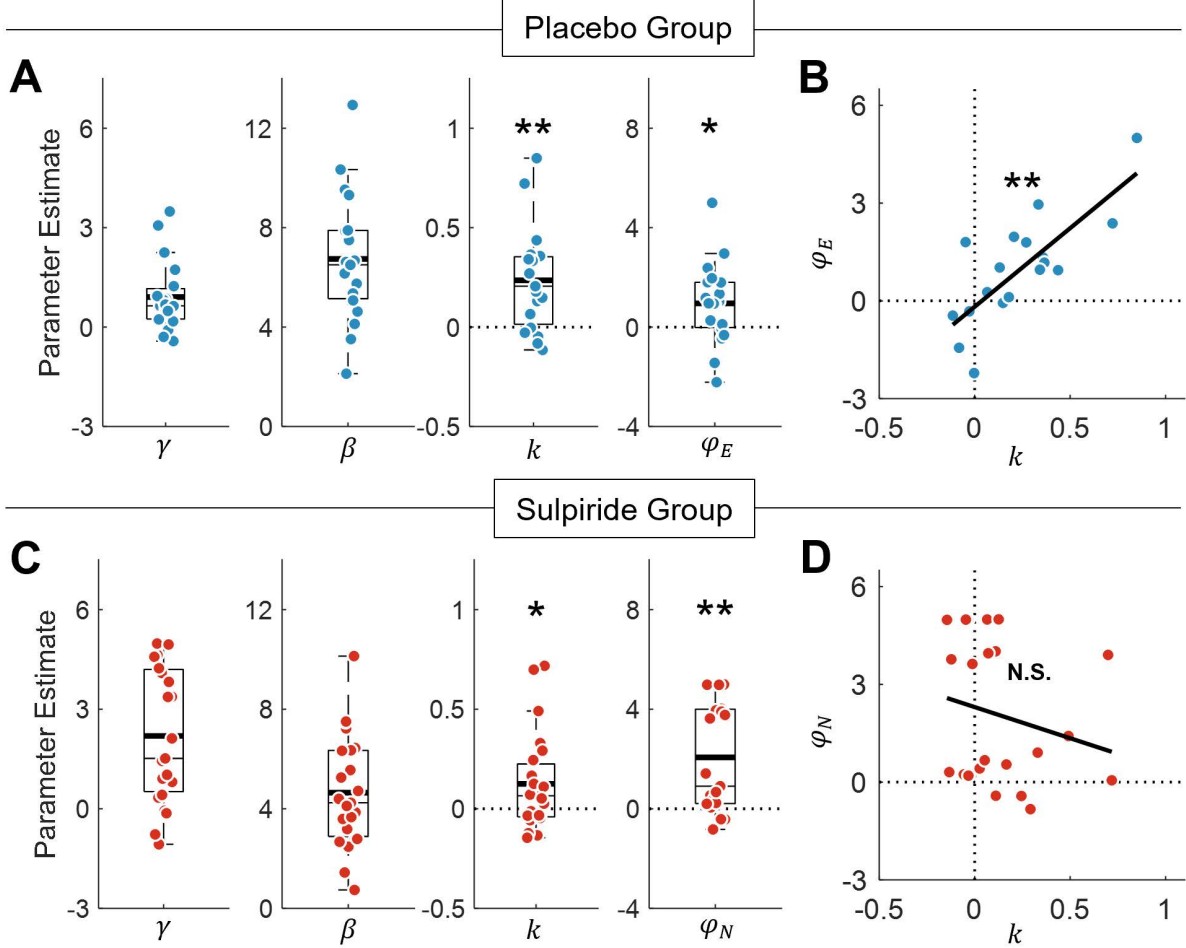

**Fig 5. Computational parameter estimates in the placebo and sulpiride groups. (A)** Parameter estimates from the *Effort Reinforcement model* (M3) in the placebo group ($n = 19$), including signal gain ($\gamma$), inverse temperature ($\beta$), effort discounting ($k$), and effort-sensitive learning rate asymmetry ($\varphi_E$). $k$ and $\varphi_E$ parameters were both significantly greater than zero, confirming that the prospect of effort was aversive ($p < .001$), and that the exertion of effort increased positive and decreased negative learning rates ($p = .019$). **(B)** $k$ and $\varphi_E$ parameters were positively correlated ($p < .001$), indicating that effort modulated learning rates to a greater extent in those more averse to exerting it. **(C)** Parameter estimates from the *Dual Learning Rates model* (M2) in the sulpiride group ($n = 23$). The $k$ parameter was significantly greater than zero ($p = .021$), consistent with effort aversion. Effort-insensitive learning rate asymmetry ($\varphi_N$) was also positive overall ($p < .001$), indicating higher positive than negative learning rates. **(D)** $k$ and $\varphi_N$ parameters were not significantly correlated in the sulpiride group ($p = .33$). Data underlying this Figure can be found in S1 Data.

that those with higher *k* values also had higher $\varphi_E$ values (r = 0.77, *p* < .001; Fig 5B). This suggests that individuals whose learning was most sensitive to effort exertion were also those least willing to choose high effort actions in the first place. We used a permutation test to verify that this significant correlation was not merely due to these parameters trading off during model fitting [64]. On each permutation, we generated synthetic data from M3 using randomly sampled, uncorrelated *k* and $\varphi_E$ values for 20 participants. Across 1,000 permutations, the recovered correlation between these parameters was as or more extreme than our empirical result on just three occasions, confirming that the relationship between *k* and $\varphi_E$ in the placebo group was most likely driven by a true correlation between effort discounting and effort reinforcement (permuted *p* = .003).

One possible interpretation of this result is that effort plays an adaptive role by facilitating learning in those who are averse to exerting it in the first place. To test this possibility, we first fit a linear regression model predicting mean accuracy in the placebo group as a function of effort aversion (*k*), learning rate asymmetry (φ), and their

interaction. Although none of these parameters was significant ($p \geq .21$), the direction of the interaction effect would be consistent with an adaptive role for effort in participants most averse to investing it (S1 Text; S5 Fig). To investigate this more thoroughly, we ran a simulation analysis testing the predictions of model M3 in four groups of $n = 100$ simulated agents (Fig 6A, 6B). Each group varied according to their degree of effort aversion ('low', $0 \leq k \leq 0.15$;

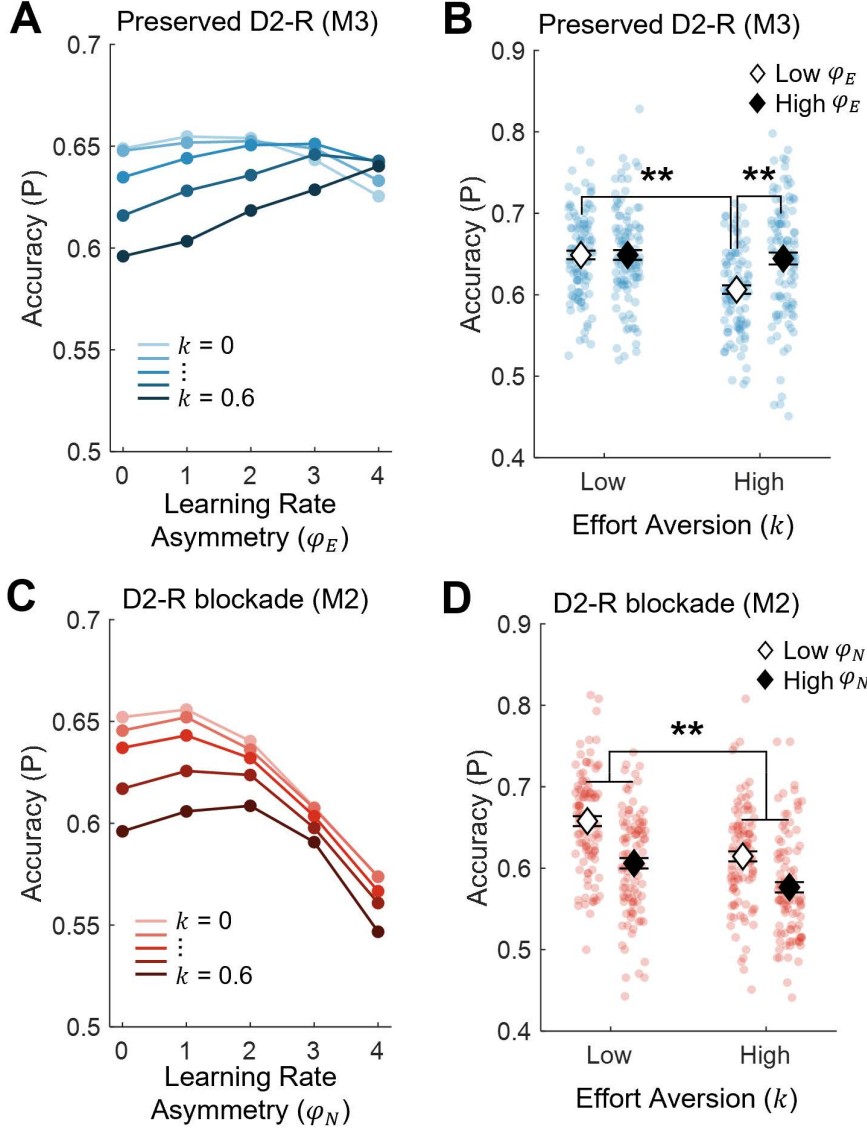

**Fig 6. Simulated learning performance under conditions of intact and disrupted dopamine signaling. (A)** Learning performance simulated using model M3 at positive values of learning asymmetry ($\varphi_E$; x-axis) and effort aversion (**k**; color shades). Each data point depicts mean choice accuracy (y-axis) based on 1,000 learning agents (total $N = 25,000$). **(B)** Choice accuracy (mean ± SEM; y-axis) of M3-simulated learning agents with low vs. high effort aversion (**k**; x-axis), further divided into those with low vs. high learning rate asymmetry ($\varphi_E$; white vs. black markers). Each group includes 100 simulated agents (total $N = 400$). The model predicts that, with preserved D2-receptor function, having more effort-sensitive learning rates improves the learning performance of individuals more averse to investing effort ($p < .001$). **(C)** Simulation of model M2 ($N = 25,000$) at positive values of learning asymmetry ($\varphi_N$; x-axis) and effort aversion (**k**; color shades). **(D)** Choice accuracy (mean ± SEM; y-axis) of M2-simulated individuals with low vs. high effort aversion (**k**; x-axis), further divided into those with low vs. high learning rate asymmetry ($\varphi_N$; white vs. black markers). Each group includes 100 simulated agents (total $N = 400$). The model predicts that following D2-receptor blockade, effort aversion is detrimental to performance irrespective of learning rate asymmetry ($p < .001$). Data underlying this Figure can be found in S1 Data.

'high', $0.45 \le k \le 0.6$), and the degree to which effort modulated their learning rate asymmetry ('low', $0 \le \varphi_E \le 1$; 'high', $3 \le \varphi_E \le 4$). Signal gain and temperature parameters were fixed at plausible values close to the observed group means ($\gamma = 2$, $\beta = 6$).

We formally tested the interaction between $k$ and $\varphi_E$ in a two-way, between-subjects ANOVA comparing overall choice accuracy. This revealed a significant $k \times \varphi_E$ interaction (F(1,396) = 10.1, $p = .002$), such that greater effort aversion (high versus low $k$) led to lower choice accuracy in the low $\varphi_E$ group ($\Delta$ Accuracy $= -0.042 \pm 0.009$, t(396) = $-4.99$, $p < .001$), but not the high $\varphi_E$ group ($p > .99$). Accordingly, in simulated agents who were most averse to investing effort (high $k$), having learning rates that were more sensitive to effort (high versus low $\varphi_E$) significantly improved learning performance ($\Delta$ Accuracy $= 0.038 \pm 0.009$, t(396) = 4.49, $p < .001$; low $k$ group, $p > .99$).

This result indicates that agents who were less motivated to invest effort (high $k$) had lower accuracy relative to those who were more motivated (low $k$). Importantly, however, this effect was only seen in those whose learning rates were less sensitive to effort (low $\varphi_E$), and not in those whose learning rates were more sensitive to effort (high $\varphi_E$). This suggests that effort-sensitive learning rates may confer a learning advantage by significantly improving performance in those who are more averse to effort, while at least maintaining performance in those who are less averse to effort.

### On sulpiride, greater effort aversion impairs accuracy regardless of learning rate

To investigate whether dopamine plays a role in maintaining this adaptive relationship between effort and learning rates, we performed the corresponding simulation on the sulpiride group using the M2 model. We began by testing whether effort discounting ($k$) in this group was significantly related to the asymmetry of learning rates ($\varphi_N$). Recall that, in this model, $\varphi_N$ is not sensitive to effort, and therefore should not be expected to correlate with effort discounting. Accordingly, this correlation was not significant ($p = .33$; Fig 5D), confirming that effort discounting did not influence learning rate asymmetry once effort exertion was decoupled from learning.

We examined the predictions of model M2 by simulating learning performance in four groups of $n = 100$ agents using the same range of parameter values as in the previous analysis. Specifically, groups had fixed signal gain ($\gamma = 2$) and inverse temperature parameters ($\beta = 6$), but varied according to their degree of effort aversion ('low', $0 \le k \le 0.15$; 'high', $0.45 \le k \le 0.6$), and learning rate asymmetry ('low', $0 \le \varphi_N \le 1$; 'high', $3 \le \varphi_N \le 4$; Fig 6C, 6D). A two-way ANOVA showed significant main effects of both $k$ and $\varphi_N$. In particular, accuracy was significantly lower in agents who were more versus less averse to effort ($\Delta$ Accuracy $= -0.036 \pm 0.006$, F(1,396) = 34.03, $p < .001$). In addition, accuracy was significantly lower in those exhibiting large versus small learning rate asymmetries ($\Delta$ Accuracy $= -0.045 \pm 0.006$, F(1,396) = 51.72, $p < .001$). Importantly, the $k \times \varphi_N$ interaction was not significant ($p = .27$), indicating that learning performance was impaired in agents who were less motivated to invest effort, and this detrimental effect of low motivation was seen regardless of the degree of learning rate asymmetry.

Together with the preceding simulations, these results reveal that D2-receptor blockade removes a mechanism that can compensate for decrements in performance that otherwise result from being less motivated to exert effort.

## Discussion

This study demonstrates a causal role for dopamine in supporting the interaction between effort and learning. In the placebo group, more forceful motor responses resulted in more efficient learning from positive outcomes, and less efficient learning from negative outcomes. Critically, blocking dopaminergic transmission with sulpiride disrupted the effect of effort on learning rates, and resulted in poorer learning accuracy relative to placebo. Model simulations revealed that effort-sensitive learning rates may play an adaptive role in maintaining learning performance. Under placebo, the capacity of effort to modulate learning rates served to, at the very least, maintain performance regardless of an agent's level of motivation. In contrast, under sulpiride, agents who were less motivated to exert effort had poorer overall accuracy, and learning rates (that were no longer sensitive to effort) could no longer offset this deficit. Together, these data demonstrate an adaptive role for effort in modulating learning rates, and reveal a novel function of dopamine in maintaining this critical relationship.

The roles of dopamine in effort and learning are often studied in isolation. Here, we examined the interaction between effort and learning using a novel reward learning task which required individuals to integrate the learned value of a prospective reward with the effort required to obtain it. We then fit computational models that captured the effect of effort on learning at the level of single trials. Across both the placebo and sulpiride groups, learning rates were higher following positive relative to negative outcomes, which is consistent with previous reinforcement learning studies [38–41]. Importantly, in the placebo group, the exertion of effort modulated this learning rate asymmetry by further enhancing learning from positive outcomes, and attenuating learning from negative outcomes. This represents an important replication of previous work [41], and demonstrates the robustness of the finding that, in the presence of preserved dopaminergic signaling, effort reinforces learning.

Critically, we found that sulpiride disrupted the relationship between effort exertion and reward learning, which indicates that dopamine plays a central role in coupling these two processes. This finding broadly contrasts with traditional models of dopaminergic signaling, which draw a clear distinction between the role of dopamine in generating effortful actions versus learning from reward outcomes. For example, the tonic activity that guides engagement in effortful behavior [65,66] is typically distinguished from the phasic activity that drives reward-based learning [35,36]. However, more recent data suggest that such a dichotomy may oversimplify a more complex relationship [15–17]. For example, effort has also been associated with transient spikes in dopaminergic activity [47,67–69], and this phasic activity appears to play a role in reward valuation [15,47,70]. These recent findings predict that dopamine may serve to mediate an effect of effort exertion on reward processing. Here, we provide direct causal evidence in favor of this interpretation in humans, by showing that dopaminergic blockade dissolves the link between effort and reward learning. This result presents the possibility that transient, effort-induced dopamine signals may act on post-synaptic D2-receptors [71] to modulate phasic learning rates, and that this effect may be disrupted by D2 antagonism. However, the precise neurophysiological correlate of our effect will need to be determined in future studies.

Our findings raise the teleological question of why learning rates should be sensitive to effort at all. Data from the placebo group provide some insight into this question. On placebo, effort had a greater effect on learning ($\varphi_E$) in those who were more averse to exerting it ($k$; Fig 5B). These data parallel the psychological concept of 'effort justification', which describes the tendency of individuals who are more averse to investing effort to overvalue the rewards of any such investment [72]. This tendency is often attributed to the cognitive dissonance that arises from the aversiveness of effort [73]. However, our simulations suggest that the effect of effort on learned reward values may confer a behavioral advantage. In particular, agents who were more averse to the prospect of effort achieved a higher choice accuracy if their learning rates were more versus less sensitive to effort exertion itself (Fig 6B). This suggests that the coupling of effort and learning may represent an ecological mechanism [74] that mitigates against potential performance decrements associated with low motivation.

The consequences of dopaminergic blockade in our study highlight the functional importance of a mechanistic link between effort and learning. Dopamine has long been implicated in the regulation of learning rates [6–8,75,76]. Thus, our finding that overall accuracy was lower on sulpiride versus placebo was in itself unsurprising. However, recent work has also shown that dopamine encodes reward information in the context of effortful behavior [47,48]. Our model simulations shed light on the behavioral consequences of decoupling effort and reward through dopaminergic blockade. In the sulpiride group, accuracy was poorer in learning agents who were less motivated to exert effort relative to those who were more motivated (Fig 6D). Importantly, unlike in the placebo group, this detrimental effect of being poorly motivated could not be mitigated by higher learning rate asymmetries—which, in this group, were no longer sensitive to effort. Our simulations thus point to an adaptive role for dopamine in maintaining learning outcomes regardless of one's motivation to exert effort.

Together, these findings provide broad insights into the functions of dopamine in learning and motivation. For example, our data build on established frameworks of reinforcement learning [36,77] by suggesting that dopamine controls learning in a specific, directional manner as a function of the effort requirements of candidate actions. The current results also inform an influential model of motivation, which stipulates that striatal dopamine encodes the average reward rate of

the environment, and that the vigour of an action should increase when more rewards are available [65,66,78]. Here, we describe a mechanism that might facilitate this process—namely, that the exertion of an effortful action enables individuals to adapt to, and learn more quickly about, any increases in the prevailing reward rate.

An intriguing question is whether the observed effects of effort should hold in contexts other than reward-based learning. For example, recent work has proposed that choice behavior is shaped not only by rewards, but by the formation of habits via direct strengthening of recently taken actions [79–81]. It seems plausible that the dopaminergic mechanisms underpinning habit formation might also be sensitive to the amount of effort with which these actions are executed [82–84], but to our knowledge this has not been directly tested. It is also unclear whether our results should generalize to learning from punishment—that is, outcomes that are not only worse than expected, but wholly detrimental to the decision-maker. Given the complex role of dopamine (and other systems) in punishment learning [85–87], it is difficult to predict the specific modulatory effects of effort in this domain. Our results thus present interesting directions for future work on the role of effort outside the context of reward learning.

Finally, these results have important implications for clinical disorders of dopaminergic dysfunction, in which motivational deficits are often accompanied by learning deficits, such as in Parkinson's disease (PD) [6], ADHD [88], and schizophrenia [89]. For example, individuals with PD often exhibit altered effort-based decision-making [20,90–93], as well as reinforcement learning [6–13]. Our data suggest that, in addition to impairments in motivation and learning, the behavior of these patients may be further compromised because effort is no longer able to 'rescue' the impaired learning that results from greater effort aversion. Future work will be needed to test this prediction in clinical groups with dopaminergic dysfunction.

In summary, we demonstrate a novel function of dopamine in supporting an adaptive link between effort and reinforcement learning. Our data show that learning rates are sensitive to effort exertion, and that the dopamine D2-receptor plays a causal role in maintaining this effect. By modeling the effects of effort and reward-based learning within a common computational framework, this work advances earlier attempts to reconcile the role of dopamine in effort and learning across species [15,94–96], and invites further consideration of how effort-learning interactions drive motivated behavior in health and disease.

## Supporting information

**S1 Text. Supplementary analyses.** Full details of additional statistical and computational analyses, including: session effects on choice behavior; drug effects on heart rate and blood pressure; drug effects on subjective feelings; assessment of participant blinding; controlling for effects of drowsiness; controlling for effects of effort variability; testing differences between experimental blocks; testing the effect of sulpiride on choice exploration; exploring the interaction between $k$ and $\varphi$; computational models with declining learning rates; computational models fit across both drug groups.
(DOCX)

**S1 Data. Processed data underlying main analyses.** Excel workbook containing processed data underlying the main statistical analyses in the paper, including data depicted in Figs 3A–3G, 4A–4E, 5A, 5C, 6B, 6D, S1A–S1F, S2, S3A, S3B, and S4.
(XLSX)

**S1 Fig. Session effects on behavior.** PL, placebo; SP, sulpiride. Total $N = 42$. Error bars depict the standard error of the mean. Numbers inside markers in panels E and F denote first and second sessions. *$p < .05$, **$p < .01$. Underlying data can be found in S1 Data.
(TIF)

**S2 Fig. Drug effects on heart rate and blood pressure.** Effects are plotted as a function of time post-ingestion of the sulpiride (red, $n = 23$) or placebo (blue, $n = 19$) capsule. Error bars depict the standard error of the mean. bpm, beats per minute; sBP, systolic blood pressure; dBP, diastolic blood pressure. Underlying data can be found in S1 Data.
(TIF)

**S3 Fig. Drug effects on subjective feelings.** Effects are plotted as a function of time post-ingestion of the sulpiride (red, $n = 23$) or placebo (blue, $n = 19$) capsule. (A) Aggregated factor scores. (B) Selected individual scales. Error bars depict the standard error of the mean. Underlying data can be found in S1 Data.
(TIF)

**S4 Fig. Learning curves averaged over contingency blocks.** Accuracy (mean ± SEM; y-axis) in the sulpiride (red, $n = 23$) and placebo (blue, $n = 19$) groups on each trial since the most recent change in stimulus-reward contingencies (x-axis). Hollow markers depict accuracy during learning, solid markers depict accuracy after contingencies have been fully learned. Underlying data can be found in S1 Data.
(TIF)

**S5 Fig. Interaction between k and φ.** Exploratory simple slopes analysis showing predicted choice accuracy (y-axis) as a function of learning rate asymmetry (φ; x-axis) and effort aversion ($k$; colors) derived from M3 in the placebo group.
(TIF)

## Acknowledgments

The authors thank Dylan Curtin, Mindaugas Jurgelis, Julia Koutoulogenis, Bridgitt Shea, and Eleanor Taylor for assisting with participant recruitment and screening; Patrick Cooper and Ziarih Hawi for overseeing participant randomization; and all those who volunteered as participants in this study.

## Author contributions

**Conceptualization:** Huw Jarvis, Trevor T.-J. Chong.

**Data curation:** Huw Jarvis, Amy Q. Huynh.

**Formal analysis:** Huw Jarvis.

**Investigation:** Huw Jarvis, Oluwadamilola Obawede.

**Project administration:** Huw Jarvis, Oluwadamilola Obawede, Amy Q. Huynh.

**Resources:** James P. Coxon, Mark A. Bellgrove, Trevor T.-J. Chong.

**Supervision:** James P. Coxon, Mark A. Bellgrove, Trevor T.-J. Chong.

**Writing – original draft:** Huw Jarvis.

**Writing – review & editing:** Trevor T.-J. Chong.

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
