## [Editor Report · Decision Letter 0]

4 Aug 2025

Dear Dr Jarvis,

Thank you for submitting your manuscript entitled "Dopamine D2-receptor blockade disrupts the effect of effort on learning" for consideration as a Research Article by PLOS Biology.

Your manuscript has now been evaluated by the PLOS Biology editorial staff, as well as by an academic editor with relevant expertise, and I am writing to let you know that we would like to send your submission out for external peer review.

Once your full submission is complete, your paper will undergo a series of checks in preparation for peer review. After your manuscript has passed the checks it will be sent out for review. To provide the metadata for your submission, please Login to Editorial Manager (https://www.editorialmanager.com/pbiology) within two working days, i.e. by Aug 06 2025 11:59PM.

Kind regards,

Taylor

Taylor Hart, PhD,

Associate Editor

PLOS Biology

thart@plos.org

---

## [Decision Letter · Decision Letter 1]

16 Sep 2025

Dear Dr Jarvis,

Thank you for your patience while your manuscript "Dopamine D2-receptor blockade disrupts the effect of effort on learning" was peer-reviewed at PLOS Biology. It has now been evaluated by the PLOS Biology editors, an Academic Editor with relevant expertise, and by several independent reviewers.

In light of the reviews, which you will find at the end of this email, we would like to invite you to revise the work to thoroughly address the reviewers' reports.

As you will see, the reviewers wrote that the study is interesting and the approach is clever. However, the reviewers raised concerns about several aspects of the study, including the methodology, analysis, interpretations, textual clarity, and contextualization within the prior literature. You should carefully consider the points raised by the reviewers and address them thoroughly, especially those related to the possible behavioral impacts of drug side effects.

Given the extent of revision needed, we cannot make a decision about publication until we have seen the revised manuscript and your response to the reviewers' comments. Your revised manuscript is likely to be sent for further evaluation by all or a subset of the reviewers.

**IMPORTANT - SUBMITTING YOUR REVISION**

*Re-submission Checklist*

*Published Peer Review*

*PLOS Data Policy*

*Blot and Gel Data Policy*

Sincerely,

Taylor

Taylor Hart, PhD,

Associate Editor

PLOS Biology

thart@plos.org

REVIEWS:

Reviewer #1: Jarvis et al investigated the role of dopamine in the interplay between effort exertion and reinforcement learning. They used sulpiride to block dopaminergic signalling in a between-subject design where participants perform a probabilistic reinforcement learning task where they have to also exert the right amount of effort to obtain reward. Sulpiride impaired choice accuracy and win-stay behaviour. They fit a set of RL models and find that a model that incorporates effort exertion fits best in the placebo group (replicating their previous work), but that this is not the case for the sulpiride group. Further analysis indicates that D2 receptor blockade disrupts the relationship between motivation to exert effort and learning rate, leading to impaired performance in a reinforcement learning task.

Overall, I found the results the authors present interesting and reasonably compelling, though their conclusions are not very straightforward and therefore not the easiest to follow. In addition, there are several aspects of the reporting that are not very clear or are lacking, and including these would make the work a lot more compelling.

As the authors state, the participants in this study receive a high dose of sulpiride. Whilst this is commendable as it ensures the effects are post-synaptic, I was somewhat surprised there were no reports of side-effects at all. At this dosage, parkinsonian symptoms surely would be likely. Did the researchers only administer the BL-VAS, or did they also include ask questions more targeted to parkinsonian symptoms specifically? Relatedly, whilst the authors do not report any difference in drowsiness at any individual time point, they did find a change in trajectory which may be informative. Did the authors try explaining changes in task performance (accuracy and win-stay behaviour) with this subjective measure taken around the time point when the participants performed the task?

It is not very clear how the effectiveness of the blinding was assessed, currently just a single p-value is reported with no explanation of what was tested. It would be useful to include the blinding data in a table, and to include a standardised index like the Bang Blinding Index (Bang et al., 2004 *Control Clin Trials*). Were the researchers running the experiment also asked what condition the participant was assigned to, if so, report that data also?

The authors conclude that sulpiride did not affect motor performance as there was no group difference in mean effort generation. However, could it be that participants within the sulpiride group produced more variable effort?

I am having trouble parsing the statement that 'learning rates are sensitive to effort exertion'. Is this not just a feature of the task, i.e. you can only learn when you exert above threshold effort? It may be that I just didn't understand this, due to lack of understanding of the task.

Figure 3D-F: are the group differences across both effort conditions or is there an interaction? Could the bars be split per condition to make this more clear?

The authors show that M2 fits the sulpiride group data better than M3, which has a better fit in placebo. As the model fits were done on the two groups separately rather than including a factor that accounts for the treatment group, this makes me wonder whether there any difference in how well the models fit the data of the two groups, especially since the differences between model fits (how much better M2/3 do compared to M1) varies quite drastically between the two treatment groups. If I eyeball it from Figure 4C and F it looks to me like the absolute model fit for the placebo data looks a bit better.

Minor

The study was set up originally as a within-person experiment, but session 2 data was not further analysed due to order effects. I would still be interested to briefly hear how the authors interpret the session 1 vs session 2 effects of sulpiride, rather than just a description of the effects which is given at the moment. Somewhat related to the session effects, were there any differences between the two blocks in the first session, or an effect of trial number of effort exertion/reinforcement learning performance?

Figure 4C/F: please specify if this is single subject data, the labels ('Placebo group') are making this a bit confusing

Figure 3C-F: can the authors specify whether the bar graphs include all trials or just the 70/30 ones?

line 381: "One possible interpretation of this result is that the dopamine D2-receptor supports" this section focusses on the placebo group analysis, so this sentence seems a bit out of place here

Figure 5C caption: M3 should be M2, presumably

Reviewer #2: Here, Jarvis et al. investigated whether effort exerted when making a choice affects learning through dopamine signalling. They used a previously developed two-armed bandit task with probabilistically rewarded options that changed in blocks. Choice selection required exerting either a high or low amount of effort (through squeezing handheld dynamometers), indicated by previously learned stimuli. The authors manipulated dopamine signalling in a subset of participants through systemic blockade of D2 receptors, and found that under such conditions choice behavior was best explained by a model that updated learning rate independently of effort exerted. This was in contrast to the placebo group, for which further simulations revealed that an intact relationship between learning rate and effort might be ecologically adaptive, enabling the maintenance of high levels of accuracy even in agents averse to effort.

This work is a clever, mechanistic extension to the lab's previous findings (Jarvis et al 2022), which itself provided a neat computational unification of two disparate theories of dopamine function: as either shaping effort-based decisions, or as a teaching signal in reinforcement learning. The effects of D2R blockade are relatively robust and the authors perform a number of control analyses to confirm that the effects observed are specific to the interaction between effort and learning rate (or lack thereof). The manuscript is well written and the figures are clear and easy to interpret. Nevertheless, I have some suggestions that are largely aimed at strengthening an already strong set of results, largely centered on deeper analysis of participant behavior.

Major comments:

1. I am unsure as to whether D2R blockade affects learning per se. It did indeed reduce performance in the task by making participants less likely to choose an option that had just been rewarded. This is not necessarily a learning effect, but might instead be a result of increased exploration (indeed, inverse temperature is lower in this group as shown in fig 5). However, a key element of the behavioral paradigm used is that the probability of reward associated with each option varies independently and across blocks within a session. The effects of this manipulation are not explored anywhere in the paper, but might serve to distinguish slow learning (after block changes) from persistent exploration. More fundamentally, learning rate - and thus presumably learning rate asymmetry - should be greater around times of block switches. Is this effect seen in the placebo group, and is it disrupted in the sulpiride group - are they slower to learn the new contingencies?

2. Figure 6 exploits a simulation approach to illustrate the effects of a range of learning rate asymmetries and degrees of effort aversion on performance in the task. I appreciate that being able to do many runs of a simulation allows for greater statistical power and the ability to systematically explore the effects of a range of parameter values (inspired by the range of those from participants). But I suggest that the authors could go one step further and exploit the individual variability in parameter fits - which is large in both drug and placebo groups - to plot these same metrics but for real, not just simulated data. Although the authors do show a convincing correlation between learning rate asymmetry and effort aversion in the placebo group that is broken by D2R blockade, whether this directly has an effect on accuracy at an individual level is not currently demonstrated.

3. I understand that as this study was conducted in humans, there is a limit to the precision of the physiological manipulations that can be performed. However, some of the discussion on the role of D2 receptors could be refined to be more consistent with canonical views about their functional attributes. For example, around line 490 there is discussion around transient, effort-induced dopamine signals acting on post-synaptic D2-receptors. However, it is well established that postsynaptic D2Rs are more sensitive to tonic dopamine due to their high baseline affinity, while D2R autoreceptors (expressed presynaptically) are more sensitive to phasic release and can shape dopamine release via a feedback loop. Indeed, there is also work showing deletion of D2 autoreceptors slows reversal learning (http://dx.doi.org/10.1523/ENEURO.0229-17.2018). There is other work showing that postsynaptic D2Rs can also in fact be sensitive to phasic dopamine (https://doi.org/10.1016/j.neuron.2014.08.058) but again, my suggestion considering space and scope constraints is to at least present the canonical view (which is not made clear in the current text). As an aside, I agree with the authors that the tonic/phasic dichotomy is likely more complex than previously appreciated, but it still might hold value strictly in the context of receptor subtype activation which is of course pertinent to the current work. Finally, there is a similar issue in the introduction - beginning line 64, the first mention of D2 receptors is in the context of exerting effort. It is not that I disagree with the citations listed, but my suggestion would be to at least frame them as counter to the canonical model of D2R activation inhibiting action (as Calabresi et al is centered on).

4. This is more of an exploratory question, but I am curious as to what the authors think the limits to this effort effect on learning are? Here the participants exert a (relative to them) small amount of effort even when choosing the high effort option, such that they are always able to exert the effort required (as indicated in fig 3b, although it might be good to mention this explicitly somewhere). This is sensible to avoid confounding effects of risk. But assuming that the main results of this paper generalise to real-world scenarios, do the authors think that the effects of effort on learning continue to follow this same functional form? For example, if an individual exerts a large amount of effort and doesn't get reward, the model here suggests that, if they are very effort averse and thus have a highly asymmetrical learning rate (considering 5B), they will in fact not learn from this lack of reward and are more likely to choose to exert effort again.

Minor comments:

1. At a few points the authors call their task a 'reinforcement learning paradigm', which is not inaccurate but is very nonspecific (in principle, any task with reward and feedback can be modelled using RL). Rather, the task should probably be referred to as a two-armed bandit which is then modelled using reinforcement learning.

2. I am unsure as to how Fig. 3C is related to motor performance. As the authors point out, the probability of choosing the low effort option is matched across the two groups, but isn't the more pertinent fact that they are always able to carry out the effort required as shown in 3B? (I am assuming the dotted lines in 3B relate to the threshold for successful effort exertion). 3C seems to speak more to whether participants are effort averse.

3. Line 165-166: please be more explicit as to what a 'positively-valenced' (or negatively-valenced) auditory tone is. Is the pitch different, or something else?

---

## [Decision Letter · Decision Letter 2]

20 Mar 2026

Dear Dr Jarvis,

Thank you for your patience while we considered your revised manuscript "Dopamine D2-receptor blockade disrupts the effect of effort on learning" for publication as a Research Article at PLOS Biology. This revised version of your manuscript has been evaluated by the PLOS Biology editors, the Academic Editor, and the original reviewers.

Based on the reviews, we are likely to accept this manuscript for publication. Please also make sure to address the following data and other policy-related requests.

IMPORTANT: Please ensure that your next revision addresses the following points:

**Title:

We think that it would be helpful to include in the title that your study was conducted in humans. Is this alternate version acceptable to you?

"Dopamine D2-receptor blockade disrupts the effect of effort on learning in humans"

**Ethics:

The Ethics statement needs to be a separate, independent (and the first) subheading in the Material & Methods section. It must include any relevant the protocol/permit/project license number. Your study must also have been conducted according to the principles expressed in the Declaration of Helsinki.

**Data:

Thank you for uploading the data to Github. We require a DOI (which you could obtain through Zenodo). We also require the processed numerical data underlying the plots. Can you please add this, either in a supplementary excel file (S1 Data; "S1_Data.xlsx") or in the online repository? This applies to the following figure panels:

3ABCDEFG

4ABDE

5AC

6BD

S1ABCDE

S2

S3AB

S4

Please also include sample sizes and the meaning of error bars in all relevant figure legends, or in the figures themselves.

Please cite the location of the data clearly in all relevant main and supplementary Figure legends, e.g. “The data underlying this Figure can be found in S1 Data” or “The data underlying this Figure can be found in https://doi.org/10.5281/zenodo.XXXXX”

Please ensure that the Data Statement is finalized for publication.

**Supplement:

The current version of your manuscript includes a Supplementary Analyses document. Because supplementary documents are not proofread and are rarely examined by readers, our general preference is that these items be incorporated into the main text where feasible. But, we do permit supplements of this sort when their contents are ancillary to the main results.

If you choose to retain the supplementary document file, please double check that it is accurate because it will be published as-is.

Please also refer to all supplementary items in accordance with the scheme "SX Item" (with a corresponding filename "SX_Item.xxx"). For example:

Refer to S1 Fig (S1_Fig.tif);

Refer to S2 Fig (S2_Fig.tif);

Refer to S1 Text (S1_Text.docx); etc.

**Code availability:

Per journal policy, if you have generated any custom code or scripts during the course of this investigation, we require that you make it available without restrictions. Please ensure that the code is sufficiently well documented and reusable, and that your Data Statement in the Editorial Manager submission system accurately describes where your code can be found. Please also ensure that you choose a license for your code and include a Readme file.

We expect to receive your revised manuscript within two weeks.

*Published Peer Review History*

*Press*

Sincerely,

Taylor

Taylor Hart, PhD,

Associate Editor

thart@plos.org

PLOS Biology

REVIEWS

Reviewer #1: The authors were very responsive to mine and the other reviewer's questions and comments, with many additional analyses which have cleared up any reservations on my part. I have no further concerns and look forward to seeing this manuscript out.

Reviewer #2: I thank the authors for their thoughtful response to my comments. Their additional analyses and edits to the manuscript fully satisfy my concerns.

---

## [Editor Report · Decision Letter 3]

3 Apr 2026

Dear Dr Jarvis,

Thank you for the submission of your revised Research Article "Dopamine D2-receptor blockade in humans disrupts the effect of effort on learning" for publication in PLOS Biology. On behalf of my colleagues and the Academic Editor, Matthew Rushworth, I am pleased to say that we can in principle accept your manuscript for publication, provided you address any remaining formatting and reporting issues. These will be detailed in an email you should receive within 2-3 business days from our colleagues in the journal operations team; no action is required from you until then. Please note that we will not be able to formally accept your manuscript and schedule it for publication until you have completed any requested changes.

PRESS

Sincerely,

Taylor

Taylor Hart, PhD,

Associate Editor

PLOS Biology

thart@plos.org